# Temporal Alignment of Dual Monitor Accelerometry Recordings

**DOI:** 10.3390/s21144777

**Published:** 2021-07-13

**Authors:** Jan Christian Brønd, Natascha Holbæk Pedersen, Kristian Traberg Larsen, Anders Grøntved

**Affiliations:** Centre for Research in Childhood Health/Research Unit for Exercise Epidemiology, Department of Sports Science and Clinical Biomechanics, University of Southern Denmark, 5220 Odense, Denmark; nhpedersen@health.sdu.dk (N.H.P.); ktlarsen@health.sdu.dk (K.T.L.); agroentved@health.sdu.dk (A.G.)

**Keywords:** clock drift, method, measurement bias, sensor fusion

## Abstract

Combining accelerometry from multiple independent activity monitors worn by the same subject have gained widespread interest with the assessment of physical activity behavior. However, a difference in the real time clock accuracy of the activity monitor introduces a substantial temporal misalignment with long duration recordings which is commonly not considered. In this study, a novel method not requiring human interaction is described for the temporal alignment of triaxial acceleration measured with two independent activity monitors and evaluating the performance with the misalignment manually identified. The method was evaluated with free-living recordings using both combined wrist/hip (*n* = 9) and thigh/hip device (*n* = 30) wear locations, and descriptive data on initial offset and accumulated day 7 drift in a large-scale population-based study (*n* = 2513) were calculated. The results from the Bland–Altman analysis show good agreement between the proposed algorithm and the reference suggesting that the described method is valid for reducing the temporal misalignment and thus reduce the measurement error with aggregated data. Applying the algorithm to the *n* = 2513 samples worn for 7-days suggest a wide and substantial issue with drift over time when each subject wears two independent activity monitors.

## 1. Introduction

Objectively monitoring physical activity (PA) using activity monitors on a population-wide scale is an important element of public health surveillance and is increasingly being used in research due to concerns with inaccuracy and bias in self-reporting [1]. Traditionally, the most frequent use is a single device worn at the waist or wrist. The ability to combine measurements from multiple devices has received increased attention due to improved accuracy for the assessment of PA [2]. Moreover, recognition of common human activity types like lying, sitting, standing, walking, running and biking from accelerometry data using multiple independent monitors has been addressed in several studies [3,4]. Importantly, using multiple independent and long duration accelerometry recordings require appropriate synchronization of the temporal data collection before joint signal processing can be successfully carried out.

The activity monitors provided by ActiGraph (GT3X, GT3X-BT) and Axivity Inc. (AX3) are commonly used with the assessment of PA. These types of activity monitors are self-contained and independent in the sense that each device includes a microcontroller unit (MCU), battery, memory, various sensors (accelerometer, temperature, light and so on) and importantly a real time clock (RTC) to keep track of data collection. The battery and memory capacity of the instruments provide the option to record un-processed acceleration for 7–10 days. Combining the accelerometry recorded with multiple independent monitors require the internal time, controlled by the RTC, to be synchronized but also that the temporal data collection over time is sufficiently stable to avoid misalignment.

The temporal synchronization with data collected from multiple sensors within the same monitor is trivial as data collection from the different sensors is handled by the same MCU and thus synchronized to the same RTC. The ActiGraph GT9X Link manufactured by ActiGraph Inc. is a good example of such a device. The GT9X Link is equipped with two accelerometers, a gyroscope and a magnetometer and all data collection is handled internally by the same MCU. Data collection with multiple independent units placed at different anatomical landmarks requires synchronization either during recording or after recording has been terminated. Synchronizing measurements during recording require devices to be connected either wired or wireless. A wired connection between individual units was used with the IDEEA2 system and a wireless connection with the IDEEA3 both available from Minisun (Fresno, CA, USA) [5]. The wear comfort of a wired setup is possibly challenged with free-living recordings over multiple days or weeks. The wireless protocol between units clearly improves wear comfort but requires an increase in power consumption compared to wired communication. This reduces battery life and recording duration. The maximum recording duration available with the IDEEA3 system is 24 h, which is insufficient for measuring physical activity with epidemiological studies recommending 7–10 days of recording [6,7]. To our knowledge there is currently no research grade device commercially available for measuring human acceleration that provides sufficient recording duration and synchronization of the actual data collection.

Synchronizing the acceleration recorded with multiple independent monitors worn by the same subject is commonly performed by updating the time and date of the RTC with the time and date of the host computer. RTC devices are commonly manufactured with an accuracy of ±20 ppm (0.002%) at +25 °C and an accumulated drift of ±12 s after a 7-day recording is to be expected. Moreover, the expected accumulated drift of the different monitors is most likely not the same, which will cause the temporal synchronization of the data collection to drift as time progresses. Thus, combining the measurements of two independent devices extends the potential accumulated drift between devices to ±24 s. In a study conducted by Schuna et al. (2015), it was demonstrated that the ActiGraph GT3X real time clock accuracy contributed to an accumulated drift of −4 to 14 s over a 7-day time period [8]. The consequence of an accumulated drift of 14 s at day 7 is that none of the acceleration data samples collected for processing data windows of 2 s (used for activity type classification by Skotte et al. (2014) and Brønd et al. (2019)) are overlapping after collecting the first 24-h of data. Furthermore, as time progresses, the temporal time difference between the combined data windows will increase and this will cause algorithms to combine movements recorded with one monitor with movements recorded by the other monitor, which has been executed with a substantial time difference. The results of the time drift demonstrated in Schuna et al. (2015) were confirmed by Steel et al. (2019), who also evaluated the drift when combining accelerometry and GPS [9]. The effect of the temporal misalignment was not evaluated in these studies, and no solution proposed. The accumulated drift after 7 days estimated in the study by Schuna et al. (2015) suggests that combining acceleration collected for aggregated data of 5–10 s is not valid unless some temporal alignment is applied.

Different methods are available for the temporal alignment of noisy and independent signals [10]. With absolute alignment of signals the centroid, cross correlation, zero-phase and maximum position are methods which are commonly used. Centroid based methods are very sensitive to noise and demonstrate worse results as compared to cross correlation [10]. The zero-phase alignment method use the shift property available with the Fourier transform and the maximum position aligns the data based on the maximum value. The different alignment methods have been evaluated in a study by Gil-Pita et al. (2005) and the results suggest that the zero phase method demonstrated the best performance in terms of signal to noise ratio with high resolution radar signals [11]. However, the listed alignment methods are all valid under the assumption that the shift between the signals is constant and does not change with time. The time drift demonstrated with accelerometry recordings is not constant with time and therefore it is not valid to use the methods with complete accelerometry recordings. In a recent study by Folgado et al. (2018) a dynamic time warping method was described for the temporal alignment of acceleration and gyroscopic time series data [12]. The method was evaluated with data collected from six subjects performing the same movements but with no restriction to execution speed. The data was sub sectioned into small data blocks to only focus on a single intended movement and not the full recording. Thus, even though the movement was not executed with the same speed this preparation step of the data will greatly improve the methods ability to align the data. The dynamic time warp method aligns the data by reducing the sum of Euclidean distances between the two data series and allows for non-linear alignment of the data. The time warping method has the potential to accommodate the shift with time. However, the duration of the accelerometry recordings impose computational and storage challenges which makes the time warping infeasible on the complete accelerometry recording with data collected over multiple days. Moreover, if the temporal misalignment caused by the RTC is linear and the alignment is solved by allowing non-linear adjustment of the data, it potentially introduces local optimization to the data, which is not appropriate.

There is currently no method available to reduce the linear misalignment caused by the RTC in long duration accelerometry recordings with multiple independent activity monitors.

The aim of this study is to describe a method not requiring human interaction for the linear temporal alignment of free-living acceleration recordings using two independent activity monitors and to evaluate the method with the combined wrist and hip and also combined thigh and hip recordings over multiple consecutive days. The accuracy of the temporal alignment is evaluated by comparing the estimated initial offset and accumulated day 7 drift with a manually identified reference.

## 2. Materials and Methods

### 2.1. Participants

Eleven students, including three men and eight women, were recruited among sport science students at the University of Southern Denmark for evaluating the performance of the proposed method with wrist- and hip-worn monitors (WH group). The mean (±SD) age, height, and weight of the students were 28.9 (±6.80) years, 171.6 (±8.51) cm, and 69.8 (±10.46) kg, respectively. All participants were generally healthy. The students were approached during a class session, where they received oral information. Written information about participation was provided afterwards. The study protocol did not require registration according to a decision made by the Ethics Committee of the Region of Southern Denmark. A written informed consent was provided by all participants.

For evaluating the performance of the proposed method with thigh- and hip-worn monitors a subsample of 30 subjects were randomly selected from the PHASAR study (TH group). The PHASAR study is a school-based epidemiological study (*n* = 2674) with the purpose of evaluating the development of PA in Danish school children during a nationwide school reform with mandatory requirements of PA in school [13]. An overall summary of the temporal alignment will be presented for all subjects having more than 6 h of movement (vector magnitude acceleration >0.068 g) with both the hip- and thigh-worn monitor.

### 2.2. Measurements

In the WH group, subjects wore one monitor on their non-dominant wrist in a wristband and one monitor on the hip in an elastic band. In the TH group, subjects wore one monitor on the right leg mid-thigh and the other monitor at the hip and both monitors were attached using an elastic band. All participants in both groups were instructed to wear both monitors at all times for a 7-day period while attending their normal daily activities. The subjects were allowed to remove monitors during water activities and subjects in the WH group were also allowed to remove monitors during sleep hours.

The tri-axial accelerometer Axivity AX3 (Axivity Ltd., Newcastle, UK) was used with all measurements and placements. The Axivity AX3 measures acceleration data in gravity units (g = 9.81 m/s^2^) and in three orthogonal axes, including ambient light and temperature. The size of the unit is 23 mm × 32.5 mm × 7.6 mm and it weighs 11 g. The storage available with the Axivity AX3 monitor provides the option to record acceleration for 30 days at 12.5 Hz or 14 days at 50 Hz. A 100 Hz sampling frequency was used with the WH group and 50 Hz with the TH group. All instruments used the same ±8 g measurement range. The data were resampled to 30 Hz for all subjects using linear interpolation after download. Monitor initialization, data download and resampling were performed using the OMGui version 1.0.0.29 on the same host computer [14].

### 2.3. RTC Accuracy and Temporal Alignment

The temporal alignment and specifically the clock synchronization during data collection with wireless sensor networks (WSN) have been studied for quite some time [15,16]. With WSN each individual sensor and node in the network has its own RTC similar to devices measuring acceleration, and clock synchronization is addressed by broadcasting individual timing messages between nodes in the network [16]. The clock of each node in the network is ideally configured such that C(*t*) = *t*, where *t* stands for the ideal time. Due to the RTC accuracy the clock of the *i*th node is modeled as follows:(1)$$C_{i}\left(t\right)=\;f*t+\theta \;$$
where *θ* is the clock offset (phase difference) and *f* the clock skew (frequency drift). The clock skew or frequency drift is the arbitrary offset from the nominal frequency of the crystal oscillator used of the RTC. Clock synchronization with respect to an ideal or absolute time is not easily solved but is approached by using one node as reference. The clock relationship between two nodes, Node A and Node B, can be described as follows:(2)$$C_{B}\left(t\right)=f^{AB}*C_{A}\left(t\right)+\theta^{AB}$$
where *θ^AB^* and *f^AB^* is the relative clock offset and skew between Node A and Node B. If the clock between the nodes is perfectly synchronized then *θ^AB^* = 0 and *f^AB^* = 1. In WSN the clock synchronization can be regarded as the process of removing the effects of random delays from multiple timing message transmissions [16]. The process of broadcasting timing messages is not possible with two independent activity monitors, as they are not interconnected in a network. Thus, clock synchronization and specifically the temporal alignment have to be addressed using the acceleration per se. However, this is only possible under the assumption that the acceleration measured at two different wear locations share some similarities, as this provides the option to determine the delay or time lag at different time points during the recording from the acceleration using methods like centroid, cross correlation, zero-phase and maximum position. This method is, to some extent, similar to estimating the random delays from multiple timing messages at different time points during data collection with WSN [11]. Thus, based on Equation (2) and the ability to estimated time lag at different time points it suggests that the initial offset and drift with the acceleration measured with two independent activity monitors can be determined by solving the following model:(3)$$lag_{B}\left(i\right)=\beta_{1}*time_{A}\left(i\right)+\beta_{0}\cdots \cdots i=1,\ldots ,n$$
where *lag_B_*(*i*) is the estimated time lag (in samples) and *time_A_* is the time point in seconds and the coefficient *β*_1_ and *β*_0_ representing the drift and initial offset. The units of drift are samples*s^−1^ and initial offset is in samples. The initial offset is translated into seconds by multiplying with the sampling period (1/30~0.033s/sample). The accumulated drift at day 7 (in seconds), which is used for evaluating the performance of the proposed method is calculated as follows:(4)$$Accumulated\;drift_{Day7}=\;\beta_{1}*30^{-1}*86,400*7$$

From the initial offset and drift it is possible to resample the data and thus to reduce misalignment. Estimating the drift and initial offset from Equation (3) is valid under the assumption that the RTC accuracy during data collection with both devices is constant. RTC accuracy during recording is influenced by temperature, barometric pressure, battery voltage and aging of the crystal which provides the oscillation frequency for the RTC [17]. For both aging, barometric pressure and battery voltage it is reasonable to assume that this does not affect accuracy during data collection. For the temperature there will be a difference between devices, and this is influenced by wear location, weather conditions and activities performed. If one device is worn on the wrist and one on the thigh, we expect a larger temperature difference than if the devices were worn on the hip and thigh. We investigated the temperature difference during wear time with Axivity AX3 devices worn on the wrist and hip during a free-living condition and the maximum temperature difference range was from ±5 °C with no consistent pattern. The temperature accuracy coefficient for RTC crystals is typically −0.04 ppm/°C^2^, which translates the temperature difference into an RTC accuracy of ±1 ppm. Thus, the influence of temperature on the RTC accuracy is small and with no consistent pattern it was decided to exclude the temperature in the estimation of the initial offset and drift and simply model the association of drift with time as linear as described in Equation (3).

The accurate estimation of the time lag e.g., from cross correlation, is under the assumption that some distinct acceleration pattern caused by movement is present in the acceleration measured with both devices. Heel strike during walking, running or jumping has the potential to generate a pattern that is detectable in the acceleration measured at multiple body locations. The acceleration measured at the different wear locations is a result of the internal and external forces acting on the body. The external forces are generated by the ground reaction force, which propagates up through the lower segments. An acceleration measured at the wrist needs to propagate a longer distance than the acceleration at the hip and this might potentially introduce a delay in the signal response. The delays were investigated (results not presented) with acceleration measured at 800 Hz and video at 240 Hz during walking and running on a treadmill and the delay was negligible. Furthermore, even though the acceleration measured at different wear locations might share substantial similarities during movements like walking or running there is also the possibility that for some movements the acceleration is less similar. The estimated time lag with these movements is associated with a substantial measurement error and introduces potential outliers. These outliers will compromise the estimation of the initial offset and drift with standard ordinary least squares (OLS). There are several robust OLS methods available for addressing potential outliers, and most methods impose an explicit limit on the effect that outliers can have on the fitted regression [18]. However, the outliers introduced from estimating the time lag with movements that are not similar are considered as invalid data points rather than a poor estimate, which is optimally handled by data-dependent weighting. Therefore, in this proposed method, we iteratively exclude data points based on the standardized residual until no further outliers are identified. In the first iteration the data points with a standardized residual larger than 1 SD are excluded and the threshold for excluding data points is subsequently exponentially increased with each iteration (1, 2.7, 15.2 …).

### 2.4. Reference Temporal Alignment

The reference initial offset and accumulated day 7 drift were determined manually using visual alignment of the unprocessed 30 Hz vector magnitude acceleration data. The manual alignment process was performed by identifying the number of data samples required to align the acceleration data during the first 0–5 h of data (*offset_start_*) and data samples required to align the data at the end (6–7 days) of the recording (*offset_end_*). The *offset_start_*, *offset_end_* and the time points for the alignments (*Time_start_* and *Time_end_* in seconds) were then subsequently used to calculate the initial offset and accumulated drift at day 7 using the following equations:(5)$$drift=\frac{offset_{end}-offset_{start\;}}{Time_{end}-Time_{start}}$$
(6)$$Initial\;offset=offset_{start}-drift\;*Time_{start}$$
(7)$$Accumulated\;drift_{Day7}=drift\;*30^{-1}*86,400*7$$

The units of all offset variables in Equations (5)–(7) are in samples, seconds for all Time variables and in samples*s^−1^ for the drift. The manual identification of *offset_start_*, *offset_end_*, *Time_start_* and *Time_end_* was performed by visually inspecting the unprocessed acceleration and specifically by using the time shift function available in the audio editing software Audacity Version 2.3.3 (retrieved September 2020) [19]. Audacity is a free open-source software application for processing large amounts of audio data and provides powerful functions for visual editing and labeling of audio data and is developed by the Audacity Team (2020). The time shift function in Audacity enables the user to visually adjust the temporal alignment between data channels and, thus, an option to determine the number of samples that would align two independent data channels. The original accelerometry data recorded by the Axivity AX3 were stored in .cwa binary files, which were not directly compatible with Audacity. Subsequently, the files were made compatible with Audacity by converting them into uncompressed WAV audio files using the OMGui software. The thigh/hip and wrist/hip data were combined in a single WAV file using two channels (stereo) of vector magnitude acceleration without any post-processing. The two accelerometry channels were split into two mono channels to enable the time shift tool. The specific processes of determining the *offset_start_*, *offset_end_*, *Time_start_* and *Time_end_* were carried out in three steps. The first step involved the amplitude and temporal zoom function to focus on a small section of the data (5–20 s of data) and importantly to identify sufficient amplitude and distinct accelerations, which were present in both measurements. Finding the appropriate acceleration was performed by iteratively adjusting the zoom and temporal scrolling function to find the most optimal position for aligning the data. Sufficient amplitude and distinct acceleration are commonly observed with activities like walking, running, jumping or just bumping both monitors during mounting and unmounting. The second step involved the time-shift function, which enabled moving one channel with respect to the other channel and thus to obtain a perfect temporal alignment of the identified acceleration. The third and final step involved the selection tool, which was used to determine the number samples shifted.

### 2.5. Automated Temporal Alignment

The overall aim of the proposed method, as previously described, was to identify the initial offset and drift with time, which were used to resample the data. The method was divided into three stages and is visually presented in Figure 1.

In stage I, the acceleration for both devices was pre-processed and the time lag was determined for 1-h non-overlapping data blocks. Only data blocks with an average acceleration exceeding 0.01 g were used. The average acceleration was calculated as the mean of the absolute Euclidean Norm minus one (ENMO) [20]. The pre-processing of the acceleration prior to determining the lag included calculating ENMO, reducing digital noise by dead-band thresholding (only including acceleration larger than 0.068 g), reducing measurement and calibration error using band-pass filtering (4th order Butterworth IIR filter with 0.1 and 7 Hz cut-off frequencies) and converting all negative acceleration to positive (absolute). The cut-off frequency and order of the band-pass filter were chosen based on the recommended practice with filtering biomechanical data [21]. A zero time delay with the band-pass filtering was obtained by using the filtfilt function in Matlab [22]. The individual pre-processing steps were processed in the same order as listed. The 0.01 g 1-h data block threshold was simply selected to ensure that the data block was not just showing sedentary behavior, but contained some movement. The low threshold was also used to ensure that a sufficient number of data blocks were included in the subsequently estimation of initial offset and drift (see stage II). The time lag between signals was identified as the maximum correlation coefficient identified in the normalized correlogram generated with a cross-correlation. The xcorr cross-correlation function available in Matlab using maximum lag of 1000 sample points (33.3 s) was used to calculate the correlogram [23]. The correlogram generated with subject three from the WH group at three time points is presented in Figure 2 as an example. The three time points presented are days 1, 3 and 5. The maximum correlation coefficient (peak) clearly shifts to the right indicating the progression of the lag and specifically the drift with time. The distance of the two peaks close to the central peak for day 1 is approximately 0.5 s. This corresponds to the commonly used step frequency of 2 Hz and suggested that the central peak is the strong correlation of the movement pattern generated by foot strike during walking or running and that the side peaks are the correlation with the previous and next step.

In stage II, the offset and drift were estimated by iteratively evaluating the linear regression of the observed lag (samples) with time using the fitlm linear regression function available in Matlab [24]. As described earlier, during each iteration the standardized residual was used to exclude data points until no further outliers were identified and the threshold for excluding data points was exponentially increased with each iteration (1, 2.7, 15.2 …). The final regression analysis for subject three is presented in Figure 3 as an example. Three data points were excluded during the iterations. The individual data points are almost perfectly aligned on the regression line. The root mean squared error (RMSE) is 0.642 samples and the R^2^ value is 0.999.

In the final stage III the acceleration measured with one monitor was temporally aligned to the reference monitor using the identified initial offset and drift estimated in stage II using resampling. The reference monitor for both the WH and TH group was the hip-placed monitor. The alignment process was implemented in two steps. The first step aligned the overall data using the offset by either removing (positive offset) or prepending samples (negative offset) at the beginning of the recording. In the second step the data were resampled using the interp1 function available with Matlab [25]. Different interpolation methods are available with the interp1 function and selecting the optimal interpolation method has to be carried out with careful consideration of the subsequent use of the acceleration. In applications where the aim is to generate ENMO or the mean average deviation physical activity intensity information, which is sensitive to the frequency content of the data, it might be optimal to use the previous neighbor interpolation (‘previous’ interpolation) as this will to a large extent preserve the signal amplitude and thus frequency content. However, in applications in which the acceleration is used for activity type identification it might be optimal with linear or cubic spline interpolation (‘linear’ or ‘cubic’ interpolation). The default method used in the proposed alignment method is the ‘previous’ interpolation, but alternative interpolation methods can be used by adjusting the parameters in the function call. The Matlab code for the described method is available on Github (https://github.com/jbrond/Aligndata accessed on 1 July 2021).

### 2.6. Statistics

Bland–Altman analysis was used to assess bias and 95% limits of agreement (LOA) and a paired t-test was used to assess if the algorithm estimated initial offset and drift at day 7, which were significantly different from the manually identified references. A *p*-value < 0.05 was considered significant. All statistical analysis and data processing were performed using Matlab (Mathworks Natick, MA, USA) version 9.0.0.341360 (R2016a) and R (R Core Team, R Foundation for Statistical Computing, Vienna, Austria) version 3.6.2 [26,27]. The blandr package available with the statistical software R was used to conduct the Bland–Altman analysis [28].

## 3. Results

Initial offset and accumulated day 7 drift reference, estimated and absolute difference for each subject in the WH group are presented in Appendix A. The reference initial offset ranged from −6.2 to 3.2 s and accumulated day 7 drift from −15.4 to 22.1 s, whereas the algorithm estimated initial offset ranged from −6.2 to 3.1 and day 7 drift from −15.2 to 21.8 s. The initial offset absolute difference between reference and estimated ranged from 0.00 to 0.73 s, and day 7 drift ranged from 0.01 to 0.36 s. The algorithm estimated offset and day 7 drift were not significantly different from the references (*p* = 0.97 and *p* = 0.41). The Bland–Altman plot for the reference and algorithm estimated initial offset and accumulated day 7 drift are presented in Figure 4 for the WH group. The initial offset bias estimated with the Bland–Altman analysis was −0.016 s (LOA: −0.59 to 0.55 s) and bias for the drift at day 7 was 0.013 s (LOA: −0.51 to 0.54 s).

Initial offset and accumulated day 7 drift reference estimated and absolute difference for each subject in the TH group are presented in Appendix A. The reference initial offset ranged from −6.9 to 9.8 s and accumulated day 7 drift from −23.1 to 453.1 s, whereas for the algorithm estimated initial offset ranged from −7.1 to 9.9 s and accumulated day 7 drift from −23.0 to 441.3 s. The initial offset absolute difference between reference and estimated ranged from 0.00 to 3.21 s, and accumulated day 7 drift ranged from 0.01 to 11.78 s. The estimated initial offset and accumulated day 7 drift were not significantly different from the references (*p* = 0.86 and *p* = 0.87). The Bland–Altman plot for the reference and algorithm estimated initial offset and accumulated day 7 drift are presented in Figure 5 for the TH group. The initial offset bias estimated with the Bland–Altman analysis was 0.005 s (LOA: −4.62 to 3.95 s) and the bias for the accumulated day 7 drift was −0.33 s (LOA: −4.62 to 3.95 s). The accumulated day 7 drift estimated with subject nine was substantially longer than what can be explained by the RTC clock drift alone, suggesting subject nine as a potential outlier. The potential cause of this is addressed in the discussion, although if subject nine was excluded from the analysis, the bias for the initial offset was −0.106 s (LOA: −0.39 to 0.18 s), whereas the bias for the accumulated day 7 drift 0.06 s (LOA: −0.57 to 0.69 s).

To provide a descriptive overview of the extent of drift in a large-scale population-based study, a total of 2513 subjects from the PHASAR study database were identified with valid measurements for both thigh- and hip-worn accelerometers. The number of subjects with accumulated day 7 drift divided into six categories are presented in Figure 6. The number of subjects with less than 5 s accumulated day 7 drift was 958 (38.1%) whereas the number of subjects with substantial drift >15 s was 939 (37.4%).

The number of unique monitor serial numbers with an accumulated day 7 drift more than 30 s was 41 and only four monitors were involved in multiple recordings demonstrating the abnormal drift.

## 4. Discussion

This study is the first to describe and evaluate a method not requiring human interaction for the temporal alignment of acceleration measured by two independent monitors worn at different body placements. The visually identified reference and algorithm estimated day 7 drift in this study is comparable to the drift estimated in the study by Schuna et al. (2015) and Steel et al. (2019), with the exception of one subject in the TH group, who demonstrated an accumulated day 7 drift of 441.3 s. Despite this single subject, the present study supports the hypothesis that RTC accuracy causes the clock drift with individual monitors measuring acceleration. The identification of the initial offset and drift with time are two important elements in the temporal alignment of acceleration measured using two independent monitors. In this study, the reference initial offset and accumulated day 7 drift for all subjects were determined manually using visual inspection of the unprocessed acceleration and the results demonstrated a good agreement between the reference and the algorithm with small LOA. Moreover, the algorithm estimated initial offset and accumulated day 7 drift were not significantly different from the reference. The results also substantiate that the linear drift with time, which is an important assumption with the algorithm. The absolute difference for the initial offset and accumulated day 7 drift was marginally increased with the WH group as compared to the TH group. This might be explained by the different movements that can be expected at the wrist as compared to the hip. The arms are free to move and therefore not at all times coherent with hip movements which is valuable for describing gross body movements like walking, running and jumping. Thus, the acceleration measured at the wrist and hip is at times not in sync and this might affect the cross correlation used to assess the drift over time. However, the presented results clearly demonstrated that the proposed algorithm is valid to minimize the effect of offset and drift for combining the acceleration recorded with multiple independent monitors on aggregated data. Although, the results do not suggest that the method is valid for sample-by-sample level data.

The drift observed with one subject in the TH group is substantially larger than the <24 s expected with the accuracy of the RTC clock. A total of 25 subjects (1%) in the PHASAR data demonstrate an accumulated day 7 drift above 30 s suggesting that the results from the subject in the TH group are not a single event. The data from this subject were further scrutinized to understand the potential cause of the abnormal drift, and from the regression analysis it seems that the drift was suddenly decreased for a short moment after approximately 24-h of recording. Therefore, this is not a sudden change in the accuracy of the RTC per se but an Axivity AX3 specific behavior which seems to halt the sampling process and, thus, increase drift between monitors to a level above what could be expected based on the RTC accuracy alone. A total of 41 unique monitors were used for the recordings demonstrating an abnormal drift >30 s and only four monitors were used in multiple recordings. If the same devices demonstrate a consistent abnormal drift >30 s it could indicate some hardware malfunction. However, only four of the identified monitors were used in multiple recordings that demonstrated abnormal drift, and this suggests that the abnormal drift was more likely caused by the internal firmware and not hardware. The described algorithm is valid under the assumption that drift is linear with time, and not valid for considering the abnormal drift. We suggest that when drift is >24 s, the data should be manually investigated to ensure the validity of the alignment. However, until the issue is solved by the company (Axivity) it is possible to use the proposed algorithm to accommodate the abnormal drift to some extent.

In this study, the acceleration measurements were re-sampled to 30 Hz due to both practical and computational limitations and also to facilitate the development of an algorithm applicable with devices that do not provide long duration recordings with sampling frequencies above 30 Hz. A higher sampling frequency than 30 Hz does provide and increase temporal granularity of the data and this might improve accuracy of the temporal alignment. We conducted a post hoc Bland–Altman analysis comparing the initial offset and drift with both 30 Hz and 50 Hz sampling frequency with the WH group. The results showed a very small bias (−2.24 and 0.25 milliseconds) and narrow LOA (−0.04 to 0.035 s and −4.0 to 4.5 milliseconds) with no clear pattern indicating that an increased sampling frequency improved the accuracy with the proposed algorithm. This could indicate that the poor accuracy was caused by the wear locations per se and specifically the substantial difference in measured accelerations [29] or the temperature fluctuation during recording. However, even though increasing sampling frequency did not improve the accuracy with the proposed algorithm it might be different with alternative pre-processing or if temperature was used in modelling the lag progression with time.

An important assumption for the proposed method was the linear drift across the full measurement period. The RTC accuracy (commonly 20 ppm) was specified at 25 degrees and accuracy is influenced by temperature, barometric pressure, battery voltage and ageing of the crystal setting the oscillation frequency. This suggests that the assumption of linear drift might be compromised during measurements in a natural environment with potential large temperature fluctuations or if the subject wearing the devices dismounts one device and not the other which will introduce large systematical temperature differences. The temperature at various device wear locations during free-living measurements fluctuates with both activities performed, weather condition and wear location per se. These fluctuations in temperature may explain the reduced accuracy obtained with the proposed method. A further development of the proposed method could be introducing temperature in the prediction of offset and most importantly drift, although the difference in acceleration measured at different wear locations in combination with the slow response rate of the temperature measurements might be challenging to further improve the accuracy of the alignment method.

All monitors were initialized within minutes from each other suggesting that the initial offset was not caused by the RTC accuracy of the host computer. The offset was more likely caused by the firmware implemented in the activity monitor and specifically the execution of alternate tasks than acceleration measurements during initiation of the recordings. Moreover, during the post hoc analysis it was discovered that the OmGUI software used for resampling the acceleration data into 30 Hz and 50 Hz did not provide the exact same data in the first initial seconds of the recording. Thus, a small difference in the initial offset was introduced by the resampling process. This limitation of the software is important to address in future studies evaluating the effect of increasing or decreasing the sampling frequency with acceleration measure with Axivity monitors. The start offset between monitors might not be the same for different brands of activity monitors.

The specified ±20 ppm accuracy of the RTC seems to be the standard for most bands providing the measurement of acceleration. The manufacturing process is the source of the RTC accuracy, and devices with higher accuracy are available. However, RTC with improved accuracy including temperature compensation also increases the costs and might not remove the requirement for alignment. Matched devices with similar accuracy could be a secondary alternative, although this would complicate administration of the devices during measurements.

## 5. Conclusions

In this study, we described a method for the automated temporal alignment of triaxial acceleration measured with two independent activity monitors, which does not require the interaction by the individual. The results of the estimated initial offset and accumulated drift at day 7 demonstrate a valid method for reducing the measurement error caused by RTC accuracy when combining acceleration measured with independent devices. Moreover, applying the automated alignment method to a large cohort study including 2513 subjects suggests a wide and substantial issue with drift over time when each subject wears two independent activity monitors.

## Figures and Tables

**Figure 1 sensors-21-04777-f001:**
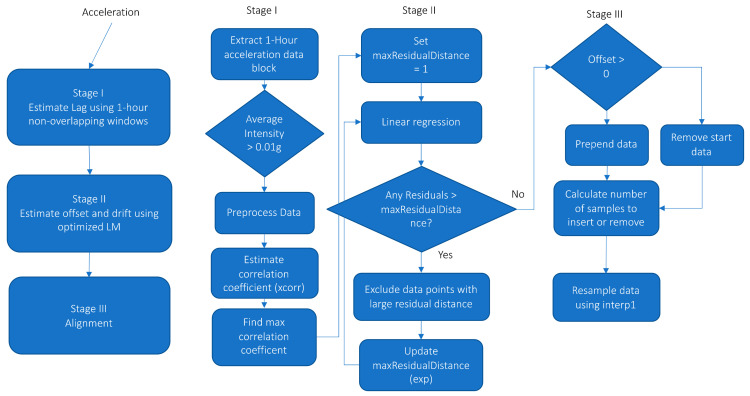
Overview of the three stages and individual processing steps used in the temporal alignment method.

**Figure 2 sensors-21-04777-f002:**
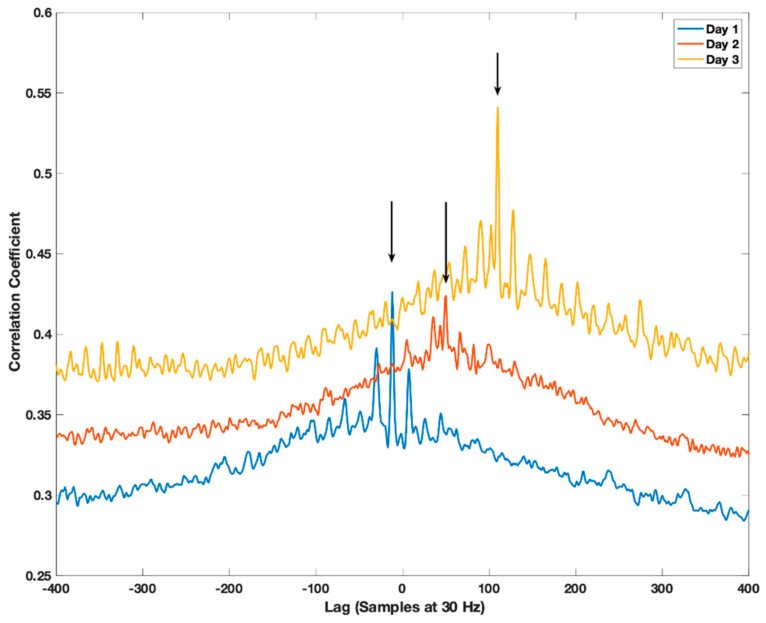
Subject three correlogram for hip and wrist pre-processed acceleration for days 1, 3 and 5. The data block for day 1 and 3 is from 02.00–03.00 PM, whereas for day 2 it is from 03.00–04.00 PM. The arrows identify the peak correlation coefficient which gives the lag and thus time shift between the two 1-h data blocks. The hip-worn monitor is the reference and thus the lag is the number of samples that is required to offset the acceleration of the wrist to align it with the hip acceleration.

**Figure 3 sensors-21-04777-f003:**
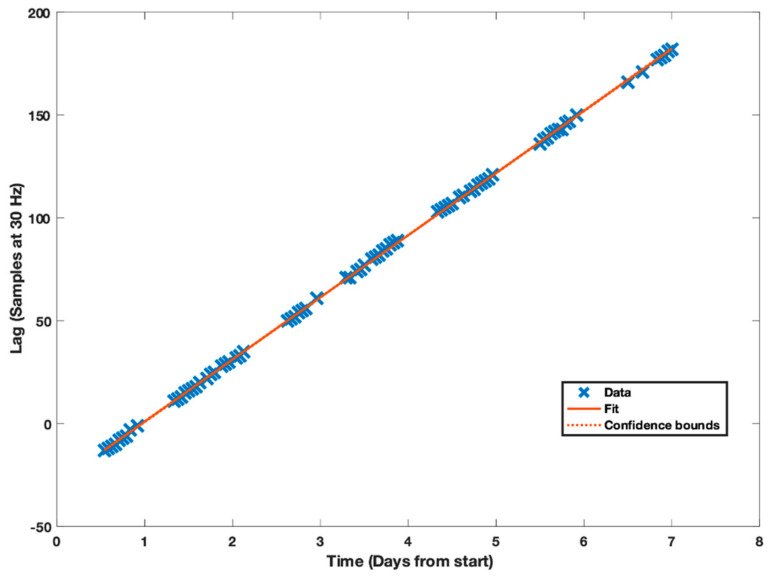
Individual data points, confidence bounds and regression line for the association between the estimated lag for the one-hour data blocks and time. This is data for subject three in the WH group. Initial offset was estimated to −12.8 samples and the drift to 30.2 samples per day. Time is the hip-worn reference monitor time and lag is the number of samples required to offset the wrist acceleration to align with the hip acceleration.

**Figure 4 sensors-21-04777-f004:**
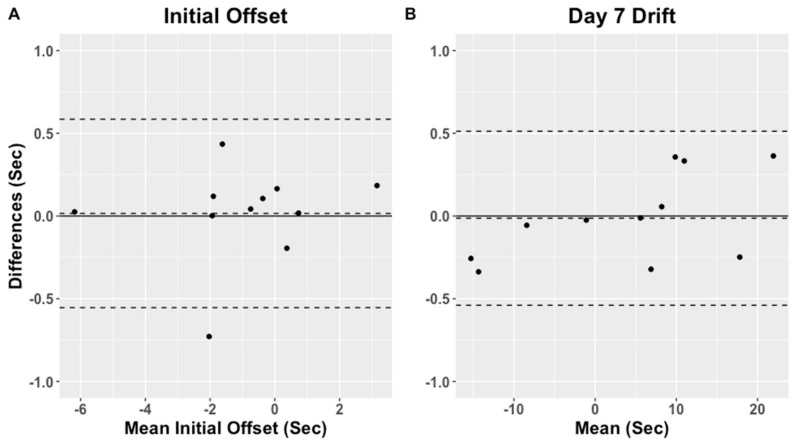
Bland-Altman plots of the initial offset (**A**) and the accumulated drift at day 7 (**B**) for the WH group.

**Figure 5 sensors-21-04777-f005:**
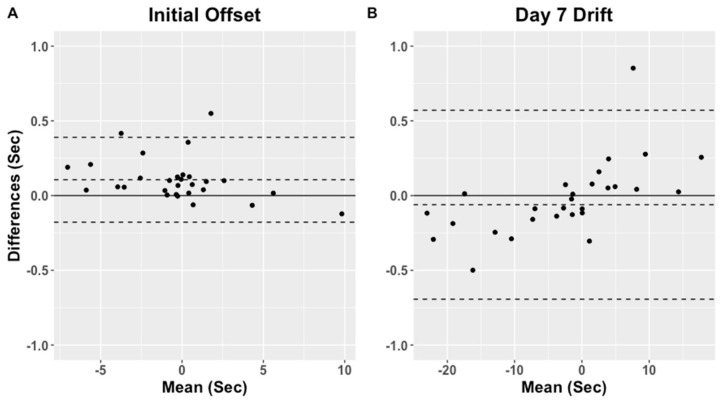
Bland-Altman plots of the initial offset (**A**) and the accumulated drift at day 7 (**B**) for the TH group. Subject nine was excluded from the plots.

**Figure 6 sensors-21-04777-f006:**
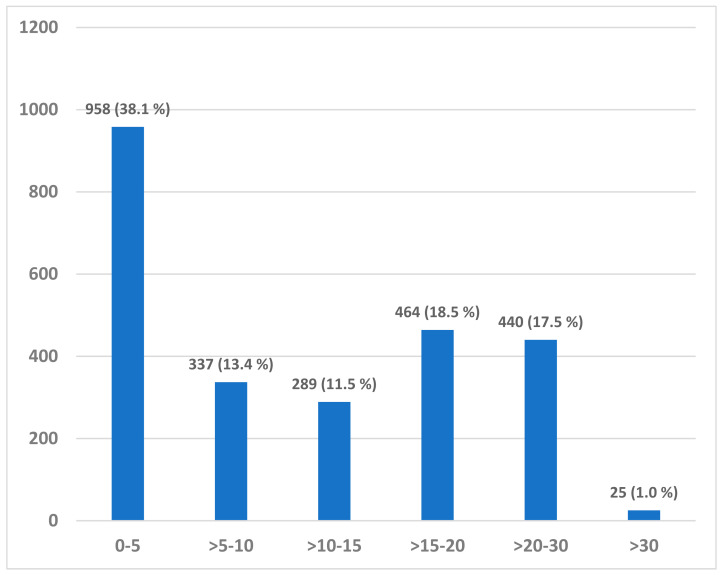
The number of subjects with an accumulated day 7 drift in seconds divided into six categories (0–5, >5–10, >10–15, >15–20, >20–30 and >30 s).

## Data Availability

The data for the WH group are available on figshare https://doi.org/10.6084/m9.figshare.12793532.v1 (accessed on 12 August 2020) and the data for the TH group are only available on request due to project and ethical restrictions.

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
