# Peer review of "Temporal Alignment of Dual Monitor Accelerometry Recordings"

_sensors, 2021, doi:10.3390/s21144777_

Round 1
Reviewer 1 Report
The manuscript is significantly improved compared to the previous version; however, several major issues remain to be resolved, especially in the newly added section about the temporal alignment of RTCs which requires substatial edits.
The remaining issues are resolvable in one additional round of reviews. Therefore, once authors correct the manuscript and once corrections are reviewed the manuscript may become ready for publication and may be accepted.
1) About the RTC model
Adding a new section on RTC alignment model significantly improved the manuscript, however there are several major issues that must be addressed.
1a) Regarding Eq. (1): I feel the word “delay” is best replaced with some better choice (also do note there may also be clock advancement instead of a delay). RTC measures time and you are in essence aligning measured times from many RTCs to one reference RTC, or in your particular case of two RTCs you are aligning one to the other.
Therefore, a more appropriate modeling equations are something like
reference time = alpha_1 * measured time + alpha_0
to align the measured times or
reference offset = beta_1 * measured offset + beta_0
to correct the offsets for non-overlapping data windows etc.
Next, note that one of the RTCs must be selected as a reference RTC. For the reference RTC the linear coefficient is per definition one (alpha_1=1, beta_1=1) and there is no offset (alpha_0=0, beta_0=0). Therefore, reference time and reference offset are actually measured values of the reference RTC while measured time is the time measured by all other RTCs in your sensor network.
This is a more precise description of your model than currently used linear regression equation.
1b) Regarding Eq. (1): Drop the epsilon_i.
There is an often overlooked key difference between statistical data analysis and model fitting.
In statistical data analysis when using linear regression we are relating two variables under the assumption of a known additive noise epsilon_i, which in this case you cannot possibly know, i.e. epsilon_i is definitely none of the standard statistical distributions. This fact is even more obvious considering all the discussion regarding temperature drifts, oscillator stability etc.
Your RTC problem is better framed in the model fitting framework where were are computing parameters alpha_1 and alpha_0 such that the fitting error (reference time - alpha_1 * measured time - alpha_0)^2 is minimized over all measurements.
Note that the actual algorithm to compute the coefficients for those two problems becomes exactly the same if one assumes the Gaussian distribution for epsilon_i, therefore there is no need to redo the experiments and reprocess the data. However, as you cannot assume the normally distributed noise for RTCs you have to reframe the problem in the model fitting framework where the fit error w.r.t. the L2 norm is minimized.
Also note that a much better robust fit would use the L1 norm, i.e. minimize abs(reference time - alpha_1 * measured time - alpha_0) instead.
1c) Regarding drifts:
RTC measure their relative time by counting periods of a known crystal oscillator. We have two different uses for the word drift, a frequency drift and a clock drift. The frequency drift is the arbitrary offset of an oscillator from the nominal frequency and it is modelled by alpha_1 (or beta_1). The clock drift is the accumulated time difference (either delay or advancement) with regard to the reference clock. You must estimate the frequency drift to resample the measurements. You are interested in clock drift which indicates the quality of your measurement device.
Please clearly introduce those two drifts in this section and verify the use of the word drift throughout the manuscript.
2) Regarding the section “Reference temporal alignment”:
Please rewrite this section considering the changes to the previous section.
Considering definitions of frequency and clock drift given under 2c) Eq. (2) does not define neither of those two drifts. If units are samples and seconds then Eq. (2) actually computes the sampling frequency using the RTC which produced Time_end and Time_start measurements as the reference etc. Please re-read and correct this section carefully so it aligns with the introduced model.
3) Regarding the section “Automated temporal alignment”
3a) The description of the algorithm is much improved compared to the previous version, but please re-read everything again as the description in Fig. 1 again references three “sections” of the alignment method while the diagram contains “stages”.
3b) Regarding band-pass filtering consider adding a literature reference (either here or in the introduction) regrading the standard frequency content of typical human movement so readers of Sensors could better understand the justification for selected cut-off frequencies, e.g. https://doi.org/10.1016/0021-9290(85)90043-0 or similar. Such a reference may also be used to provide a better insight and justification into why resampling from 100Hz to 30Hz is acceptable which you discuss in Section 4.
3c) There is no absolute time, i.e. everything is re-expressed in the time axis of an arbitrarily selected RTC which is a reference. Please stress this fact in this section. Also clarify descriptions of Figs. 2 and 3: In which time frame is the lag axis in Fig. 2, the first or the second RTC? In which RTC is time, and in which RTC is lag in Fig. 3?
3d) Starting from 292: Although the algorithm for linear regression and for L2 error minimization of the fitting error are exactly the same, as explained in 1b) it would be better to replace linear regression to model fitting where RMSE is minimized.
Other minor issues:
4) Lines 442-445 (clean version) and the response to Issue 10) in the previous review: You agreed with the assessment that the attribution of a sensor behavior to a firmware glitch is best left out unless there is a clear understanding of what is the cause. Yet the revised text reads:
“…which seems to suggest that the abnormal drift potentially was caused by the internal firmware and not nessesarly hardware. The drift caused by the internal firmware can not be expected to be constant as the drift caused by the accuracy of the RTC clock.”
As previously suggested drop this part as it is not needed. You have clearly described how such invalid measurements can be detected which is sufficient.
5) Similarly, MCU vs CPU remains in line 39 (clean version).
6) (repeated issue 12 from the previous review) In several places third person singular verb endings are incorrect etc. Full language editing is not required, but please find someone proficient in English who did not read the manuscript to help in proof reading.
The first mistake remains at lines 10-11: …a difference in the real time clock accuracy… …introduceS… and there are others.
I stress the part: “did not read the manuscript”.
Reviewer 2 Report
Figure 1 requires minor modifications, as the labels present overlapping letters and such, also a different font would help.
The manuscript requires a language check, as there are some commas out of place and minor spelling errors.
It is not clear how the automated temporal alignment was implemented, which tools were used to achieve automation of the signal post-processing.
It is mentioned that accuracy is influenced by battery voltage, temperature and ageing of the oscillator. It is implied that the affected accuracy refers to the temporal synchronization. Nevertheless, these factors could also affect the quality of the recorded signal, as in amplitude or noise, which in turn could affect an automated process. How is this adressed in the presented study?
Round 2
Reviewer 1 Report
All issues raised during the review were appropriately addressed and the manuscript may be accepted for publication. Regarding the English language (and in my opinion due to a really short time between re-reads) there are still minor language errors present, e.g. an before consonant (an post), missing definite articles, etc., so please try to find a proofreader (a colleague would do) who has not read the manuscript yet to briefly check the final version. A formal language editing is not required.This manuscript is a resubmission of an earlier submission. The following is a list of the peer review reports and author responses from that submission.
Round 1
Reviewer 1 Report
The paper presents results of automatic temporal alignment attempt for a multiple independent accelerometer recorders. The topic is important in analysis of the movement or any other activity where measurements from multiple independent devices are used.
The paper structure is suitable and the results are presented appropriately. However, the methodology is not adequate and it is also not presented in a way clear to the reader. The following are the major identified concerns and comments about the methodology:
- Sampling frequencies of the WH and TH groups were 100 Hz and 50 Hz, respectively, and then downsampled to 30 Hz. I see no good reason for the downsampling when both devices in the same group (WH or TH) have the same sampling frequency! By reducing the sampling frequeny you are also reducing the achievable alignment accuracy. For example, at downsampling from 100 Hz to 30 Hz, the corresponding sampling time rises from 10 ms to 33.33 ms and the average achievable alignment accuracy rises from 5 ms to 16.66 ms.
- I seriously doubt about the ability to perfectly align two independent accelerometer signals from two devices only by manual visual inspection, as claimed in line 181-182. The movements, especially in WH setting, are generally not synchronized and since the person doing the alignment does not know what movement it is, it is not possible to do such alignments with the needed accuracy. Much more believable scenario would be to make a standard movement at the beginning and the end of the 7 day measurement. The best would be bumping the two devices three to five times by the lab staff and then the alignment of those spikes!
- The presentation of the automated temporal alignment procedure is poor. There a several questions that remain unanswered:
- How exactly did you determine the 10 mg0 threshold? Did you take into account possible sensor biases that are larger than that? Is that threshold done on separate axes or on an absolute acceleration value?
- What do you want to achieve with dead-band threshold at 0.068 g0? Why the two thresholds then at 10 and 68 mg0?
- Why converting all negative accelerations to positive when you use (as claimed) the magnitude signal anyway?
- The preprocessing of the signals for the automated temporal alignment procedure have serious flaws:
- It is wrong to subtract the g0 value from the absolute value of acceleration (magnitude) of all three axes. For example, when the sensor Z-axis is aligned with the gravity vector and the device is accelerating with 0.5, 1 or 2 g0 in the direction of the X-axis, the respective errors by simply subtracting the gravity are 0.39, 0,69 and 0,77 g0 --> what is a substantial error.
- Why do you use the 7 Hz cut-off frequency in the band-pass filter? This reduces the (needed) signal spikes for alignment!
- There is no clear description in what order the preprocessing procedures are used,
- It seems that authors use some digital signal processing techniques without proper justification or need.
Other comment are:
- Equations on page 4 are not numbered and are very hard to read. The units of variables cannot be the part of the equation and must be given within the describing text!
- The temporal alignment procedure using those equation are not clear. How exactly do you get the offsets and start/end times using both signals? A figure picturing it would be very helpful.
- The propagation delays of the heel strike are claimed to be negligible, but you do not give the values, even though you claim you have measured them on a treadmill with 800 Hz sampling frequency!
- Figure 2 is illegible. Use two different colors!
- Figure 3 is explained poorly, the two peaks close to the central one are not distinguishable.
- There is no explanation what the Figure 4 actually mean! Is that the proof that the time drift is linear (RTC clock inaccuracy is constant) or something else. The time in the graph should be given in hours!
- In tables 1 and 2 the bottom line with averages is calculated incorrectly! Averages for the differences must be calculated on absolute values of differences. I suggest removing the average line.
Reviewer 2 Report
Feedback items:
This research presents an algorithm for aligning independently sample IMU data streams, taking into account the clock drift that can occur when samples are collected over large time periods (i.e. generally multiple days of sampling will produce significant clock drift in type RTC-based samples). The problems area is of significant interest to the areas sensing and time series analysis.
The introductory section would benefit from more literature on studies that cover/incorporate the task for aligning multiple data streams. This is a common pre-processing task that is required in many studies and the author’s needs to further emphasise the novel aspects of the proposed workflow.
Is Linear drift across all data samples a fair assumption? It would be nice to include a small section that motivates this assumption. I suspect that the patterns of drift over time will be specific to the model of the device and/or the specific unit. Battery voltages and temperatures may also affect the consistency of the RTC hardware. A justification around why the drift should be applied in a linear fashion would be nice.
The equations presented in lines 167-172 should be numbered and referenced in the explanatory text. The explanation should be expanded here to clearly cover the application of the equations – I assume these are applied to the data streams in a pair-wise fashion, but this protocol is not clear within the text.
I would expect to see the Bland-Altman plots that accompany the analysis. This appears to be missing.
The results presented in tables 1 and (especially) 2 could be presented visually and eliminate the need to randomly select a subset of the results – this gives no feel for the distribution of the results. A series of simple scatter plots or the Bland-Altman plots should provide a better feel for the distribution of the differences between the reference and estimated values.
Consider using colour in line plots to make individual data series easier to identify (colour is used in other plots – why not in the line plots?)
As mentioned before discussion in 376-397 might benefit from some background on the causes of RTC drift. Line 403-413 there is the potential for battery performance to be affected by temperature, causing voltage differences in the device (in addition to the decrease in voltage the charge dissipates). This may affect the RTC and CPU depending on the devices.
The conclusion section is brief and doesn’t link well with the stated aims of the study.
Editorial Feedback:
Lines 28-31 – reword the sentence to make sense.
Line 35 – ‘appropriately’ should read ‘appropriate’
Line 78 – ‘progress’ should read ‘progresses’
Lines 82-83 – Mixed citation format.
Line 167-172 – Include equation numbers.
Line 186 – OMGui citation needed,
Line 231 – Matlab citation for specific function documentation.
Table 2 – Use a consistent number of significant figures in reported values.
Figure 4. X-Axis should be converted to a day number from the start – not a listing of the individual day.
Reviewer 3 Report
The authors present a method for temporal alignment of accelerometer data. In its current form the manuscript is not suitable for publication as the details of the proposed method are not clearly described and as no underlying model of sensor’s time measurement is described, all of which make the work quite difficult to reproduce. As the manuscript was already revised once and as the remaining flaws require at least a major revision my recommendation is to reject the manuscript.
Major flaws:
1) Sensor/RTC model should be more in focus
As currently presented, there is no section which clearly describes the time measurement model via equations. The authors discuss RTC devices and their specifications in the Introduction (lines 68-90) where they clearly explain the problem, but nowhere in the article is a model of how performed time measurement deviate from expected local time presented. The reader may guess that the model is probably comprised of a fixed time offset and of a linear correction term to the frequency to take the drift into account, however this can only be indirectly inferred from Eqs. (1)-(3) and from the description in subsection “Automated temporal alignment”. The title of the journal is Sensors and at least one subsection should be devoted to explaining what the time measurement model is and how it is applied to align the data.
2) Description of the alignment method must be improved
Given the diagram of the method in Fig. 1 and the accompanying description many details are left open: 2a) Data preprocessing per description uses ENMO, dead-band thresholding, BP IIR Butterworth filtering, and taking absolute value. All stated operations are probably sequential, but this is not explicitly stated. Then, filtering introduces time delay (group delay of a filter) - is this time delay accounted for and how? Finally, per Fig. 1 preprocessing is done after average intensity detection, but from line 264 I conclude that ENMO computation is always performed first and in essence all operations in Fig. 1 are on ENMO signals. 2b) “More data” action is confusing as per the description in lines 308-326 you process all non-overlapping windows. 2c) The last two operations of “prepending data”/”removing start data” and “Remove single samples data”/”Insert single sample data” are particularly poorly explained. If you use non-overlapping windows than for all windows expect the first and the last one the best strategy would be to reposition the window per computed parameters and then to resample the signal taking the frequency drift into account. 2d) The provided link to code (line 342) points to an almost empty repository with last commit on Aug 12, 2020. 2e) Finally, if the model of the time axis transformation is affine, i.e. t_true = a * t_measured + b, wouldn’t it be better first to estimate the value of a, then resample all measurements using standard resampling techniques from signal processing such as e.g. Matlab’s resample, and only then estimate the offset b?
To conclude on 2), current description of the proposed method makes reproduction of the work quite difficult which is not acceptable.
Minor issues (in no particular order):
3) Lines 39, 48-52: Consider using microcontroller/MCU instead of CPU as activity monitors do not contain a classical desktop CPU.
4) Regarding Eqs. (1) and (2): The result of Eq. (3) is better named total error at one week as it is drift multiplied by time, i.e. name is not matched to the unit of measurement.
5) Lines 218-236: This description is best merged into a new section which describes used time measurement model (as suggested under 1.).
6) Fig 1.: Divide the proposed procedure into “Stages” instead of “Sections”. Sections and subsections are parts of the manuscript. Writing “In Section X” confuses the readers as they cannot be certain if the reference is to a section of the manuscript or to a “section” of the proposed procedure.
7) Pseudo-unit mg: Per SI system mg refers to milligrams, and not to milligravitational units. Instructions for authors clearly state: “SI Units (International System of Units) should be used. Imperial, US customary and other units should be converted to SI units whenever possible.” Therefore, do not use mg as a non-standard shorthand for milligravitational units.
8) Fig. 5 right: Drift or offset? Line 331 stats that drift is about 30 samples per day and Fig 5. and line 378 use seconds for drift? Please clearly define what is offset and what is drift.
9) Section 3. Results: The initial offset is of secondary interest for alignment, i.e. the initial offset is a (start) synchronization issue which may be solved in practice by simultaneously banging two accelerometers against any hard surface before the measurement stars and providing an easily identifiable starting signal (see e.g. how a clapperboard works). Of primary interest are estimated drifts and one-week offsets which should be discussed first.
10) Lines 441-462: Do not guess or speculate, i.e. the attribution of a sensor behavior to a firmware glitch is best left out unless you have a clear understanding of what is the cause. Simply describe the difference between the outlier measurement and the standard ones and clearly state how such invalid measurements can be detected.
11) Fig 1., line 262: For clarity select one of either (sliding) window or data-block.
12) In several places third person singular verb endings are incorrect etc. Full language editing is not required, but please find someone proficient in English who did not read the manuscript to help in proof reading.
Note that this review is based on the following materials only: manuscript 1159018 v2, manuscript 1159018 v1, and a supplementary document containing two tables with timing data.
Round 2
Reviewer 1 Report
The revised version has addresses some of the given comments, but not all of them and not the ones I consider important.
Since the main and only topic of this paper is the temporal alignment of signals from multiple devices worn for days, the time measurement granularity, which depends on the sampling frequency, is very important. Therefore, decimating the sampling frequency of signals after download is an unnecessary and disadvantageous procedure. The reasoning about larger need of storage and processing power AFTER DOWNLOAD does not seem a relevant choice! In the presented setup a 50 Hz would be a much better choice!
In pursuing the main goal - temporal alignment - there is no need for data processing such as dead-band threshold or ENMO. The intended use of signal for activity detection and quantification after temporal alignment is not the topic of this paper and should be left open for other possible uses.